# Computational Modeling of Extrasynaptic NMDA Receptors: Insights into Dendritic Signal Amplification Mechanisms

**DOI:** 10.3390/ijms25084235

**Published:** 2024-04-11

**Authors:** Mark Makarov, Michele Papa, Eduard Korkotian

**Affiliations:** 1Department of Neurobiology, Weizmann Institute of Science, Rehovot 7610001, Israel; 2Department of Mental and Physical Health and Preventive Medicine, University of Campania “Luigi Vanvitelli”, 81100 Caserta, Italy

**Keywords:** signal amplification, extrasynaptic NMDA receptor clusters, signal amplification, dendrite model, NR2B

## Abstract

Dendritic structures play a pivotal role in the computational processes occurring within neurons. Signal propagation along dendrites relies on both passive conduction and active processes related to voltage-dependent ion channels. Among these channels, extrasynaptic N-methyl-D-aspartate channels (exNMDA) emerge as a significant contributor. Prior studies have mainly concentrated on interactions between synapses and nearby exNMDA (100 nm–10 µm from synapse), activated by presynaptic membrane glutamate. This study concentrates on the correlation between synaptic inputs and distal exNMDA (>100 µm), organized in clusters that function as signal amplifiers. Employing a computational model of a dendrite, we elucidate the mechanism underlying signal amplification in exNMDA clusters. Our findings underscore the pivotal role of the optimal spatial positioning of the NMDA cluster in determining signal amplification efficiency. Additionally, we demonstrate that exNMDA subunits characterized by a large conduction decay constant. Specifically, NR2B subunits exhibit enhanced effectiveness in signal amplification compared to subunits with steeper conduction decay. This investigation extends our understanding of dendritic computational processes by emphasizing the significance of distant exNMDA clusters as potent signal amplifiers. The implications of our computational model shed light on the spatial considerations and subunit characteristics that govern the efficiency of signal amplification in dendritic structures, offering valuable insights for future studies in neurobiology and computational neuroscience.

## 1. Introduction

The initiation of excitation within a neuron commences at the postsynaptic membrane of an asymmetric synapse, orchestrated by the collaborative activity of α-amino-3-hydroxy-5-methyl-4-isoxazolepropionic acid (AMPA) and N-methyl-D-aspartate (NMDA) receptor channels (hereinafter, for brevity, called “channels” or “receptors”), the joint gating of which by a ligand leads to the initiation of an excitatory postsynaptic potential (EPSP). AMPA channels induce a potent and brief depolarization of the postsynaptic membrane through pronounced conductance (7–8 pS, 3–20 pS) [1,2], rapid activation ($\tau_{rise}=0.05-0.6\;ms$), and equally expeditious deactivation ($\tau_{decay}=0.3-6\;ms$) [3,4]. The resulting AMPA-channel-induced depolarization facilitates NMDA receptor activation by alleviating Mg^2+^ blocks. Distinct from AMPA-, NMDA-channels exhibit a comparatively slower conduction decay constant (19–112 ms) [5,6], influencing the duration of postsynaptic membrane depolarization. NMDA channels are composed of diverse subunits, each imparting varying conduction attenuation constants. The culmination of cellular excitation, manifesting as action potential (AP) generation, is contingent upon the efficacy of signal transmission mediated by synaptic AMPA- and NMDA-receptors. In contrast to the axon, dendritic conduction relies on passive propagation. Consequently, a pivotal determinant of conduction is the distance from inputs to the site of AP initiation. In specific neurons, such as those in Layer 5 of the neocortex, this distance may extend up to 1 mm [7,8]. At such distances, individual signals contribute minimally to action potential generation during passive conduction, with primary contributions arising from synapses proximal to the soma. Early models posited that long dendritic synapses regulated basal membrane potential in the soma [9,10,11,12]. The identification of active dendritic components has expanded the functional scope of long dendrites in signal transmission to the soma. Active dendritic components play a crucial role in the computational function of dendrites across various neuron types. The repertoire of active components may vary among different dendritic locations, orientations and diameters. Thus, certain types of voltage-gated calcium (Ca^2+^) and sodium channels, as well as small- and large-conductance Ca^2+^-gated potassium channels, have been proposed as important “players” influencing passive signal propagation along dendrites. It has also been suggested that certain voltage-dependent mechanisms are involved in the scarring of synapses in order to balance their “synaptic weight” depending on the distance to the soma/axon initial segment. Despite their very broad functional spectrum and additional means of modulation, a common feature of voltage-gated channels of all types is their fundamental dependence on membrane potential. In this context, NMDA receptors occupy a unique niche since a shift in membrane potential from rest does not itself initiate their conductance. Similarly, the presence of a ligand is a necessary but not sufficient condition for the initiation of NMDA receptor current, which occurs only when these two factors are combined. In the presence of an additional source of glutamate, this makes extrasynaptic NMDA receptor (exNMDA) clusters unique regulators of dendritic conduction, the role of which in this sense remains underestimated. It should be especially emphasized that they only modulate the conduction of individual and/or “selected” signals, without canceling spatiotemporal summation, as the fundamental basis of the integrative functions of the dendritic tree as such. 

One extensively researched aspect of active component involvement is dendritic spikes, notably observed in pyramidal cells [13,14] and as well as Purkinje cells [15,16]. Dendritic spikes can arise from various channels, including voltage-dependent sodium channels [17,18,19], Ca^2+^-voltage-dependent channels [20,21], and NMDA receptors [22,23]. These events result from the coordinated activity of numerous synapses (10–50 inputs) [24,25,26], while individual events should not be amplified by active components to prevent an increase in noise weight in the overall signaling and to preserve the correct synaptic scaling. Nevertheless, activating a substantial number of synapses might not be energetically advantageous, particularly for maintaining stable neural connections. A potential resolution to this dilemma could involve the amplification of single or small numbers of excitatory postsynaptic potentials (EPSPs) through extrasynaptic NMDA (exNMDA) receptors located in numerous clusters along dendritic shafts rather than shuttling into the synaptic receptor pool [27]. Morphologically exNMDA is defined as NMDA receptors located at least 100 nm from the postsynaptic density (PSD) [28,29]. Electrophysiologically, exNMDA exhibits unresponsiveness to low-frequency stimulation (<0.05 Hz), in contrast to synaptic NMDA [30,31,32]. Additionally, exNMDA interfaces with external glutamate concentrations, diverging from the isolation characteristic of synaptic NMDA [33]. Detection of exNMDA is achieved through preembedding immunoperoxidase and postembedding immunogold electron microscopy as well as through fluorescence light microscopy. The determination of cluster sizes remains intricate and subject to debate [34]. In a study by Petralia R. S. et al. utilizing preembedding immunoperoxidase, only approximate cluster sizes were computed. For instance, clusters of exNMDA with the NR2B subunit averaged a length of 186 nm (51–602 nm), while the diameter of an individual NMDA receptor is approximately 20 nm [28]. The functionality of exNMDA exhibits nuanced variability contingent upon spatial localization, subunit composition, and additional contributing factors. Notably, exNMDA distribution extends across both the outer membrane of dendritic spines and the surface spanning the entirety of dendritic and somatic regions [35,36,37]. For instance, within the soma, exNMDA serves to attenuate superfluous signals during the genesis of plateau potentials [38]. Its proximity to synaptic inputs further engenders suprathreshold summation of signals [38], governing plasticity [33,39], and extrasynaptic inhibition [40,41]. Despite the prevailing consensus positing NR2B as the predominant subunit of exNMDA [35,37], the elucidation of specific subunits associated with distinct functionalities within this receptor pool remains an unresolved query. While significant strides have been made in experimentally delineating the properties and functions of exNMDA, certain facets of its operation elude empirical investigation. This challenge is multifaceted, stemming from the intricate and multifarious nature of exNMDA’s role and the predominant reliance on slice preparations for experimental inquiry. It is imperative to acknowledge that neuronal networks in brain slices lack many inputs, thus potentially engendering disparate activity manifestations compared to in vivo conditions. Computational models offer a complementary avenue, affording the capacity to anticipate certain receptor properties and guide subsequent experimental investigations based on model-derived predictions. However, it is noteworthy that the literature documenting the inclusion of exNMDA within computational frameworks is presently limited. Some modelling works are discussed in Section 4.4. 

The prevailing body of research has predominantly focused on the role of exNMDA in close proximity to synaptic inputs. Within the present study, we introduce a novel inquiry into the dynamics governing the interaction between synaptic inputs and exNMDA clusters situated at a considerable distance from inputs. Employing a computational model featuring exNMDA clusters, we systematically investigate the intricacies of signal amplification. Our primary objectives include discerning the optimal spatial arrangement of exNMDA clusters concerning inputs and soma for efficient signal propagation, elucidating the advantageous attributes of subunits with a protracted time constant, and delineating the spatiotemporal requisites for the efficacious amplification of input signals through exNMDA clusters. This inquiry addresses hitherto unexplored dimensions of exNMDA functionality, shedding light on critical determinants of signal processing within neural networks.

## 2. Results

### 2.1. Separate and Combined Effects of Synaptic and Extrasynaptic Inputs

Our model demonstrates that a unitary synaptic input generates an EPSP with an amplitude close to the equilibrium potential (depolarization from −65 to about −5 mV) due to the effective activation of the entire cluster of postsynaptic receptors—20 AMPA- and 10 NMDA-types—as well as high input impedance. At the same time, AMPA receptors have a relatively short decay time constant, about 1 ms. The time constant for the extinction of synaptic NMDA receptors is much higher, on the order of 10–20 ms, but their contribution is very small. The model of a single EPSP (Figure 1A) is quite plausible in its parameters. In particular, it corresponds to the experimental data [42,43]. As follows from the simulation results, despite the high local amplitude, the signal rapidly decays according to the spatial constant. Thus, at a distance of 800–1000 μm from the entry zone, the signal decreases to negligible values (about 0.1–0.2 mV) and cannot lead to any noticeable shift in the potential in the far edge of the dendrite, where soma and axon initial segment would begin in a case of real pyramidal cell.

Introducing a cluster of approximately 80 units of exNMDA into the model (Figure 1B) at a distance of 400 μm from the EPSP source also does not lead to a noticeable shift in the Vm of the “soma” at resting Vm (−65 mV). Henceforth, we will use “soma” to refer to the most distant edge of the x axis. A slight depolarization of about 1 mV can only be observed 5–7 ms after exNMDA activation, since they can have a much slower decay time constant, on the order of 50 ms and slower [44].

However, the combination of EPSP and exNMDA initiation results in a much larger net effect (Figure 1C), where at 6 ms, the depolarization reaches approximately 2 mV. 

The combination of three sequential EPSPs arriving every 1.3 ms to separate but neighboring postsynaptic areas at a distance of few micrometers from each other does not change much the synaptic impact on soma (Figure 2A). Overall, the effect of several (up to four) combined EPSPs has minimal impact and shows little variation among them (Figure 2C). However, when the same EPSPs are co-activated with exNMDA the impact is much more pronounced, reaching up to 4 mV at the distance of 1000 µm from the source (Figure 2B,D). 

### 2.2. Effect of Latency between Synaptic and Extra-Synaptic Inputs

Figure 1 and Figure 2 represent the simultaneous initiation of EPSPs and exNMDA cluster by glutamate. In the model, we assume that exNMDA are activated at certain time point by a delivered portion of ligand and co-activated by approaching depolarization, originated from EPSPs. Another notable aspect is the time delay between the two with EPSPs preceding exNMDA activation and vice versa. In addition to the simultaneous delivery of the ligand to both inputs (blue traces on Figure 3B,D), we have checked several latencies, including 3, 5, 7, and 9 ms with implementation of exNMDA preceding EPSPs and vice versa for three sequential pulses (as in Figure 2A,B). 

Figure 3A represents a case when exNMDA begins 5 ms before the first EPSP. Panel B indicates that with longer delay, smaller summation is observed. Moreover, the decrease is not linear, being relatively small for 3 ms and very high for 9 ms. The dotted line reflects a case of the absence of synaptic input. 

When EPSPs arise 5 ms before exNMDA activation, the result of desynchronization is less pronounced (Figure 3C). Overall, the delay by 3 or even 5 ms, despite having little effect on the amplitude, is associated with much sharper rise, due to which the possibilities for further spatiotemporal summation look more favorable in this case than in the previous scenario (Figure 3D). The dotted line here represents no exNMDA activation. 

### 2.3. Analysis of the Influence of Synaptic Stimulation Frequency

To analyze how the rate of the three synaptic stimuli train affects the efficiency of interaction with exNMDA, we tested the following particular intervals: short (0.3 ms), medium (1.3 ms), and long (3 ms). The results of this test are presented in Figure 4. Panel A shows the spatiotemporal distribution of potentials at high frequency. Particularly, the dynamics of Vm at the maximum distance from EPSP sources is represented by the gray curve in panel D of Figure 4. The observed effect is most rapid in its rise and decay time course. Nevertheless, its magnitude is close to the case of the largest interval rate of EPSP: 3 ms (black curve and Figure 4C). Importantly, the kinetics of response at soma are much slower, with a peak time appearing some 2.5 ms later than for the previous case. The greatest amplitude effect was achieved in the case of an intermediate interval lasting 1.3 ms (Figure 4B,D, red curve), while its kinetics were only slightly longer than for the gray curve, with about 0.5 ms peak time difference. 

### 2.4. Effect of Spatiotemporal Distribution of EPSPs on the Interaction with exNMDA

The distribution of three successive EPSPs both temporally and spatially along with the free combination of these parameters opens up wide opportunities for testing the model under a variety of conditions. In previous cases, we considered a distance of just few micrometers between the inputs (Figure 2, Figure 3 and Figure 4). Here, from the entire variety of combinations, for simplicity, we have selected a relatively straightforward scenario, when the EPSPs are spatially separated by the distance of 100 μm. Under this condition, we consider the simultaneous firing of all three inputs (Figure 5A) and four time intervals between the inputs: 0.3 ms (not shown separately, gray trace in panel D), 1.3 ms (panel B and red trace in panel D), 2 ms (not shown separately, brown trace in D), and 3 ms (C and black trace in D). From the data presented, it appears that the greatest amplitude effect is achieved at an interval of 3 ms despite the fact that when the inputs are located nearby, the best effect is achieved at an interval of 1.3 ms (refer to Figure 4D). One may also pay attention to the altered shape of the responses and their much slower, two-component rise and decay time courses. This was particularly invisible with “dense” inputs, as shown in Figure 4. 

### 2.5. The Optimal Location of exNMDA

From the model perspective, an important issue is to determine the optimal spatial localization of exNMDA in relation to the three EPSP sources. To simplify the interpretation of the data, we have chosen the option of more “densely” sequenced EPSPs with an interval of 0.3 ms. This approach provides a better understanding of input interactions with exNMDA, located either closer to the inputs or, conversely, closer to the soma. The simulation results are presented in Figure 6. We considered several spatial locations of exNMDA: at a distance of 200 μm from the inputs (Figure 6A); 300 µm from the inputs (Figure 6B; gray trace in D); 400 µm from the inputs (Figure 6C; red trace in D); and 700 µm (black trace in Figure 6D). It can be seen that although the proximity of exNMDA to the EPSP produces the strongest local effect, it barely reaches the soma (Figure 6A). Conversely, excessive distance from the inputs (700 μm), although it creates conditions for the greatest influence on the soma, does not allow exNMDA to be sufficiently activated in terms of surrounding membrane potential. The optimal position for influencing the soma was found to be approximately an intermediate position of exNMDA, when they are located at a distance of 2/5 from the synapses and 3/5 from the edge (red trace in Figure 6D).

### 2.6. Effect of exNMDA Decay Time Constant on the Efficiency of Interaction with EPSPs

As discussed above, exNMDAs may have longer decay time constants compared to synaptic NMDA receptors due to the subunit composition of each individual channel. To determine the possible role of this insufficiently studied factor, we considered the “standard model” of successive EPSP inputs with an interval of 1.3 ms and exNMDA localization at a distance of 400 μm from them but with a decay time constant ranging from 20 ms (as in a synapse) to 100 ms (Figure 7). Panel A and gray trace in D correspond to 10 ms; B and brown trace—to 20 ms; red trace—to 50 ms (this configuration is also shown on Figure 4B); and finally, C and the black trace on D reflect the 100 ms decay time constant. Quite expectedly, an increase in the decay constant leads to an enhanced response in soma, while the nonlinear form of this dependence was somewhat unexpected. Thus, an increase in the time constant from 10 to 20 ms and from 20 to 50 ms had a much greater impact than an increase from 50 to 100 ms. Overall, it can be assumed that the functions of synaptic and extrasynaptic NMDA receptors may differ significantly. While the former are targeted at the influx of Ca^2+^ ions and further activation of various pathways of synaptic plasticity [45,46], the latter may be directly related to the regulation of the delivery of synaptic signals to the soma [38,47], which will be discussed below.

## 3. Discussion

### 3.1. Integrative Role of Dendritic Computation: Beyond Passive Conduction

The integrative and computational role of the dendritic tree is among the most important functions of many, although not all, types of dendrites. The earlier postulate about the exclusively passive role of dendrites in conduction was later revised and replaced by a model of synaptic and dendritic compartments that interact with each other on the basis of mixed, passive, and active mechanisms [12]. While many neuronal events—such as axonal transmission and backpropagation action potentials—are all-or-none and self-sustaining, the transmission of synaptic inputs rarely requires such cellular determinism. Rather, on the contrary, only competitive and integrative conduction is subject to fine modulation and depends on the spatiotemporal pattern of various inputs. In this context, the view of NMDA receptors as exclusively sources of Ca^2+^ and initiators of long-term rearrangements seems fair but limited. It is quite obvious that NMDA receptors conduct not only calcium ions but also sodium and in this sense can make a full contribution to changes in membrane potential. On the other hand, AMPA receptors play the role of the main driver of depolarization in the postsynaptic membrane. However, this does not apply to extrasynaptic NMDA receptors. In the presence of a ligand, they are able to act independently, influencing synaptic signals passing in a given time window (a matter of milliseconds) and in a given location. Of course, this requires fine synchronization of conduction with the supply of glutamate. In our opinion, a candidate for this role may be astroglia, which controls both synapses and fairly distant extrasynaptic areas, especially considering the large size of many astrocytes, their complex three-dimensional spatial configuration, as well as interastrocytic gap junctions through which Ca^2+^ signals can be transmitted between individual cells. It is also important to note that extrasynaptic NMDA receptors cannot serve as simple sources of replenishment of their synaptic pool, as has been demonstrated experimentally [27].

In the examination of dendritic models featuring singular and multiple inputs, our investigation establishes that exNMDA clusters assume an amplifying function in passively conducting signals along the dendritic structure (Figure 8A). The exploration of the quantity of exNMDA clusters on dendritic shafts and their dynamic behavior is reserved for forthcoming inquiries. Notably, the strategic placement of a limited number of clusters relative to inputs and the soma is of paramount significance. Our model, although currently accommodating only a singular cluster, reveals the existence of an optimal cluster position conducive to the most efficient signal amplification. In contrast to synapses, where NMDA receptors play a pivotal role in plasticity processes, exNMDA clusters serve a distinct function, necessitating a subunit composition aligned with this specialized role. This investigation elucidates the rationale behind the preference for subunits characterized by a slow decay constant in clustered exNMDAs.

### 3.2. The Origin of Glutamate for Activating exNMDA Amplification

The origin of glutamate and/or its agonists responsible for activating exNMDA clusters remains unknown. It is established, however, that dendritic regions housing exNMDA clusters exhibit heightened contact with axons (50%), dendrites (25%), and glia (25%) [13]. The potential activation scenarios posit a connection between clusters and axonal boutons, with simultaneous activation of inputs and exNMDA clusters. This simultaneous activation, a potential parameter for learning, may transpire either through direct axonal contact or through intermediary connections in scenarios involving distinct neurons. Alternatively, an activation scenario could involve the selection of the optimal cluster based on its distance from the soma and inputs through mechanisms reminiscent of learning. Notwithstanding, astrocytes emerge as the plausible source of glutamate for exNMDA, given their established role in synchronizing processes across synapses and dendrites, including those involving exNMDA [48,49,50,51]. Ashhad and Narayanan (2016) demonstrated the influence of astrocytes on exNMDA through direct stimulation with I3P [36]. However, the integrative communication of synapses and exNMDA by astrocytes (Figure 8B) has not been proven and remains to be investigated.

### 3.3. Spatial Organization of exNMDA Amplifiers: Clustered versus Diffused Distribution

Furthermore, the spatial organization of exNMDA, whether in clusters or diffusely distributed, bears significance. A diffuse arrangement suggests a potential reduction in the localized efficacy of glutamate signaling and reuptake, stemming from the dispersion of glutamate along the dendrite. Given the diminished density of astrocytic leaflets surrounding dendrites compared to synapses, and structure of the perineuronal nets [52], it is conceivable that glutamate distribution along the dendrite occurs with minimal mechanical impedance and that reuptake transpires with reduced force [53]. Conversely, the clustering of exNMDA enhances control over the conduction zone by locally regulating glutamate concentration.

### 3.4. Methodological Challenges in Investigating exNMDA Mediated Signal Amplification

Exploratory investigations into the phenomenon of signal amplification facilitated by exNMDA clusters are subject to various methodological limitations. In primary neuronal culture, the measurement of individual amplitudes of single EPSP at the soma offers a viable approach. However, discerning the extent of exNMDA augmentation necessitates the selective inhibition of exNMDA clusters through the localized application of blockers at distinct dendritic sites distal from the inputs. It should be noted that the localized administration of a non-selective NMDA receptor inhibitor, such as DL-2-amino-5-phosphonopentanoic acid (APV), does not guarantee the exclusive blockade of synaptic NMDA receptors, thus complicating interpretations. The premise that exNMDA clusters primarily comprise subunits characterized by slower decay kinetics may inform the selection of specific blockers. For instance, ifenprodil, an NMDA receptor antagonist, exhibits specificity towards NR2B subunits [36,54]. Selective blockade of GluN2C/D-containing NMDA receptors is achievable through the use of DQP-1105 [36,55,56]. However, conventional neuronal culture systems are unsuitable for investigating the influence of exNMDA clusters due to the absence of such clusters therein. It is worth recalling that exNMDA clusters have thus far exclusively been identified in slice cultures [28]. Nonetheless, the utilization of slice cultures precludes the possibility of localized application of NMDA blockers. Consequently, examining the impact of exNMDA on current amplitudes using NMDA receptor inhibitors fails to elucidate inquiries regarding the localization of exNMDA amplifiers. 

Due to these and other limitations, many studies of exNMDA funvtions have been conducted using computer simulations. The application of computational models has not only proven effective in clarifying deviations from established functionalities but also in exploring innovative paradigms. Wade, J. J. et al. employed a computational model to scrutinize the role of slow inward currents (SIC) induced by exNMDA in determining the reciprocal interactions between astrocytes and neurons during the process of learning. The model posits that astrocytes, through the mediation of slow inward currents (SIC) facilitated by exNMDA, possess the capacity to convey learning signals to distant synaptic loci during spike-time-dependent plasticity (STDP) by orchestrating neuronal synchronization [48]. However, existing models integrating exNMDA primarily focus on the interaction between synapses and neighboring exNMDA, activated by glutamate released from the presynaptic terminal (glutamate spillover). Singh, P. et al. elucidated the modulation of sensitivity and temporal characteristics of desensitization of the NR2B subunit, outlining the role of extrasynaptic NR1/NR2B-NMDA receptors located on dendritic spines in detecting recurrent interval bursts within a network [57]. The spatial proximity of exNMDA to active synapses has engendered scholarly interest in probing the impact of such NMDA receptors on conduction dynamics and synaptic plasticity processes. Liang, J. et al. simulated alterations in Ca^2+^ dynamics under conditions of Alzheimer’s disease. Modeling outcomes underscored that the elevation in ambient glutamate, resulting from reuptake inhibition, contributes to the perturbation of Ca^2+^ dynamics and subsequent synaptic plasticity disruptions [58]. In a parallel model, Flanagan, B. et al. illustrated how surplus glutamate potentiated SICs, subsequently compromising the fidelity of synaptic transmission [59]. Certain manifestations of exNMDA within the perisynaptic zone appear unrelated to SIC but may stem from secondary influences of intruding Ca^2+^. Consequently, exNMDA co-localized in proximity to Ca^2+^-activated potassium channels may participate in eliciting a rebound response in incoming potassium, thereby inhibiting neuronal excitability [41]. Furthermore, exNMDA assumes a pivotal role in the genesis of plateau potentials [60,61], thereby establishing a milieu conducive to heightened states of neuronal excitability [60].

### 3.5. Hypothetical Role of exNMDA Clusters in Neurodegenerative Disorders

Extensive investigation has been dedicated to elucidating the involvement of exNMDA proximal to excitable synapses in the pathogenesis of neurodegenerative disorders. The hyperactivity of these exNMDA is commonly linked with mitochondrial impairment and subsequent cell death [29,62,63]. Similarly, exNMDA cluster may serve as a factor in the onset of neurodegenerative disorders. Analogous to the diverse subunits constituting the synaptic NMDA receptor complexes [64,65], exNMDA clusters may exhibit heterogeneity in subunit composition. Alongside canonical NR2B, NR2D, and NR2A subunits, NR3A subunits may also be present within these clusters, potentially conferring neuroprotection against excessive calcium influx. Notwithstanding our model’s projection of NMDA clusters enriched with subunits featuring prolonged decay kinetics, such as NR2B, it warrants consideration that an altered subunit ratio favoring NR2B could culminate in heightened calcium influx and thereby precipitate excitotoxicity and cell death [66]. Analogously, overstimulation of exNMDA clusters may engender analogous repercussions [62,67].

Our model predicts that the exNMDA gain mechanism is sensitive to factors influencing the temporal alignment of synaptic events with exNMDA activation. Perturbations leading to temporal shifts in exNMDA cluster activation are anticipated to dampen the amplitude of ensuing signals (see Figure 3B,D). If astrocytes function as synchronizing entities, disruptions in the exNMDA amplifier could stem from various dysfunctions in astrocytic processes. These disruptions may impede the perception of synaptic signals, thereby affecting the timely activation of the exNMDA cluster via Ca^2+^ propagation. Such disturbances could hypothetically arise from alterations in the normal functioning of proteins associated with the astrocytic cytoskeleton and cell-cell contact proteins [68]. The specific pathological mechanisms directly contributing to the attenuation of Ca^2+^ wave velocity remain elusive. It is possible that delays in the initiation of Ca^2+^ waves or in the interastrocytic transmission may impact their propagation speed. The initiation of Ca^2+^ waves within astrocytic processes is contingent upon inositol trisphosphate (IP3), as its stimulation triggers a self-sustaining release of Ca^2+^ from internal stores. Certain mutations, such as knockdown of GPR78, have been observed to reduce Ca^2+^ release from intracellular stores [69], hypothetically resulting in the suppression, weakening, or elevation of the threshold for initiation of Ca^2+^ waves in response to excitatory stimuli. The knockout of the gene Cx43, encoding for the gap junction protein connexin 43, has been associated with a reduction in Ca^2+^ wave velocity, spread efficacy, and amplitude [70]. 

## 4. Materials and Methods

### 4.1. Modeling of Synaptic and Extrasynaptic Events

Our study is based on a plausible model of excitatory synaptic inputs, postsynaptic currents (EPSCs) and excitatory postsynaptic potentials (EPSPs). The primary analytical framework and tools of our model are derived from theoretical approaches and formulas that underlie many similar models of synaptic conductance. For an overview, refer to [71,72,73,74,75]. Postsynaptic excitation includes sequential activation of AMPA and NMDA receptors. The model operates step-by-step with a temporal resolution of 10 μs. 

### 4.2. Synaptic AMPA Receptors

The maximum total AMPA receptor conductance $G_{AMPA}(max)$ is 300 pS, which corresponds to the activity of about 15–20 single AMPA receptors, with $g_{AMPA}≅15-20\;pS$. Considering the limited probability of channel opening (with p≈0.8), the actual maximum total conductance is 250 pS and has a maximum total AMPA receptor current of approximately 16 pA. The time constant for the rise of AMPA receptor conductance ($\tau_{rise}$, i.e., increase to 63%) is set to 0.07 ms, and the decay time constant ($\tau_{decay}$, i.e., decrease by 63%) is 0.7 ms. 

Following these rise and decay time constants, the probability of AMPA receptor opening, and thus the conductance through channels at any given time point, may be described by the following difference of exponentials: (1)$$G_{AMPA}\left(t\right)=G_{AMPA}\left(max\right)\times \left(e^{-\left(\frac{t}{\tau_{decay}}\right)}-e^{-\left(\frac{t}{\tau_{rise}}\right)}\right),$$
where the model is based on the fundamental property of AMPA receptors to undergo rapid inactivation and desensitization along with complete removal of the excitatory neurotransmitter (glutamate) from the synaptic cleft within 1–1.5 ms after its release [76]. The resting membrane potential $V_{m}$ is set to −65 mV. 

Synaptic current, initiated by AMPA receptor conductance can be calculated using the following expression, representing the Ohm’s law for time-dependent conductance and the actual driving force:(2)$$I_{AMPA}\left(t\right)=G_{AMPA}\left(t\right)\times \left[V_{m}\left(t-1\right)-V_{eq}\right],$$
where synaptic equilibrium (reversal) potential $V_{eq}$ for glutamatergic synapses does not exceed 0 mV [77]. 

In this case, the local shift of the membrane potential induced by the AMPA receptor current follows the exponential law, based on the limited area of the postsynaptic membrane, about 2 μm^2^, which corresponds to the area of the average dendritic spine. The size of this area directly affects the leakage current through the potassium channel complex. The leakage conductivity in the model is set to 20 pS. This parameter determines the input resistance of this small membrane domain, calculated to be 50 GΩ, see for review: [78]. We simplify the model by disregarding the specific features of the different types of leakage channels, considering only their collective impact on the synaptic/dendritic decay time constant ($\tau_{m})$ set to 15 ms. Leakage equilibrium potential $E_{L}$ is selected to be equal to the resting membrane potential: $E_{L}=V_{m}=-65\;mV$.

However, during the initial stages of depolarization development, the decay time constant is analytically calculated since leakage actually depends on the membrane potential. Thus, the leakage current amplifies as the membrane becomes more depolarized. Based on this, the membrane resistance $r_{m}$ is defined as the reciprocal of the current leakage conductance. As repolarization proceeds, the effective or dynamic time constant $\tau_{m}(t)$ gradually approaches the steady state and typical value of $\tau_{m}=15\;ms$ [71,79]:(3)$$\tau_{m}(t)=\frac{\tau_{m}}{{1+r}_{m}\times G_{AMPA}\left(t\right)}$$

### 4.3. Synaptic NMDA Receptors

Unlike AMPA receptors, the NMDA receptor is both ligand and voltage dependent. According to this condition, even in the presence of a ligand (glutamate), the magnesium plug prevents flow through the NMDA receptor until it is displaced or pushed out by a depolarization wave of the adjacent membrane, initiated by both AMPA receptors and the activated fraction of NMDA receptors. Thus, properties of synaptic NMDA receptors are such that their conductivity increases rapidly as the membrane depolarizes, caused by the current through both fractions of active receptors. 

Voltage-sensitivity of NMDA receptor describes the fraction of currently available i.e., unblocked receptors. It depends on the kinetic equilibrium $K_{eq}$ between the ${Mg}^{2+}$ free and ${Mg}^{2+}$-blocked NMDA receptors. The equilibrium constant for this model has been taken as 3.6 mM. The factor that sets the voltage sensitivity of ${Mg}^{2+}$-block/unblock is described by $V_{m}(t)$ divided by empiric voltage constant $V_{C}$ of 0.062 mV [71,80]. Finally, the extracellular magnesium concentration ${\left[{Mg}^{2+}\right]}_{ex}$ is considered to be 1.2 mM. Based on these parameters, the voltage-sensitive conductance of NMDA receptor $\delta (V_{m})$ could be described by the following equation:(4)$$\delta \left(V_{m},\;t\right)=\frac{K_{eq}}{K_{eq}+e^{-\left(\frac{V_{m}\left(t\right)}{V_{c}}\right)}\times [{Mg}^{2+}{]}_{ex}}$$

The kinetics of NMDA receptors are slower, being 3 ms for $\tau_{rise}$ and 20 ms for $\tau_{decay}$. The total maximum conductance of synaptic NMDA receptors is set at 600 pS, equivalent to the conductances of approximately 12 fully open channels [81]. The final time-dependent NMDA receptor conductance $G_{NMDA}\;(t)$ equation looks as follows: (5)$$G_{NMDA}\left(t\right)=G_{NMDA}\left(max\right)\times \left(e^{-\left(\frac{t}{\tau_{decay}}\right)}-e^{-\left(\frac{t}{\tau_{rise}}\right)}\right)\times \delta \left(V_{m},t\right)$$

Therefore, the actual NMDA receptor conductivity turns out to be significantly lower than 600 pS, not exceeding 80 pS with a total NMDA receptor current of about 6 pA. 

### 4.4. Computation of EPSP

As mentioned above, the calculation of the NMDA receptor current is carried out in a step-by-step manner to account for the effect caused by both AMPA and NMDA receptor conductance. However, during ongoing synaptic event, the total AMPA and AMDA receptor currents (EPSCs) at the postsynaptic membrane do not exceed 16 pA. Based on the actual current values, the model calculates the actual membrane potential of the postsynaptic region. These time-dependent voltage measurements of EPSP(t) during synaptic event are described as follows:(6)$${∆V}_{m}\left(t+1\right)=-(V_{m}\left(t\right)-E_{L})-\left[r_{m}\times g_{EPSP}\left(t\right)\times {(V}_{m}\left(t\right)-E_{eq})\times \left(1-e^{-\left(\frac{t}{\tau_{m}\left(t\right)}\right)}\right)-E_{L}\right]$$

The simulation conditions are configured to prevent the membrane potential from reaching 0 mV (reversal potential for both AMPA and NMDA channels). This constraint also applies to sequential and/or neighboring synaptic inputs (in this model—up to four), separated in time and/or space to varying degrees. Their calculation is structured in such a way that the actual membrane potential at a given synaptic zone before its activation serves the base for farther computation. Therefore, the process of spatiotemporal summation of several identical inputs occurs considering the maximum possible membrane depolarization of 0 mV.

### 4.5. Extra-Synaptic NMDA Receptors

Basic properties of extrasynaptic NMDA receptors (exNMDA) are equal to those located at synapse. The sole significant difference lies in the subunit composition, which will be discussed later, leading to a considerably slower decay time constant of up to 50–100 ms. The total conductance has been chosen about 7 times higher than at the synapse. It corresponds to a cluster of approximately 80 exNMDA. We allocate all 80 receptors at a single spot locus. The distribution of receptors along a certain limited surface does not affect the simulation result. On its own, the basal activation of exNMDA at the resting potential only induces a minor depolarization. A shift in the membrane potential in the extrasynaptic region can only be driven by synaptic currents propagating along the dendritic shaft.

### 4.6. Propagation of Signals along Dendrite

The achieved synaptic current $i_{syn}$ values where $i_{syn}=i_{AMPA}+i_{NMDA}$ approach the dendrite, where they are able to propagate as axial currents, $i_{a}=i_{syn}$, in accordance with the solution of the cable equation for a dendrite of a certain diameter and under the assumption that there are no physical boundaries on the left and right. In this model, as a reasonable compromise, we assumed the model dendrite radius to be 0.6 µm. 

Obviously, the diameter of a real dendrite does not remain constant through its length from the apical end to the soma. However, we did not aim to analyze the effect of dendritic lumen, instead limiting the model to a single specific radius. Perhaps we will consider the influence of dendritic lumen on the interaction with extra-synaptic receptors in the future. Based on the above, we set the axial resistance of the dendrite using a resistivity $R_{a}=100\;\Omega \times cm$. The only deviation from the standard description of electrical conduction along a cable was the consideration of input impedance, which, we believe, plays an important role in the analysis of electrical events in a long dendrite [78]. Thus, the computation was carried out for a 1 mm long dendrite. Impedance was calculated taking into account a “typical” synaptic input frequency of about 75–150 Hz. Spatial resolution was based on a step of x=0.1 μm, and temporal resolution (as for synaptic events) on a step of t=10 μs. The space constant λ for dendrite was assumed to be 400 µm with $r_{\lambda }=1.85\;$GΩ. 

The original cable equation for infinite cable has the following form:(7)$$\tau_{m}\frac{\partial V}{\partial t}=\lambda_{m}^{2}\frac{\partial^{2}V}{\partial x^{2}}-V+r_{m}\times i_{a},$$
and has the following solution [71,75]:(8)$$V\left(x,t\right)=\frac{i_{a}(t)\cdot r_{m}}{\sqrt{2\cdot \pi \cdot \lambda_{m}^{2}\cdot t/\tau_{m}(t)}}\times e^{-\frac{{\tau_{m}\cdot x}^{2}}{4\cdot \lambda_{m}^{2}\cdot t}}\times e^{-\frac{t}{\tau_{m}}}$$

The current flowing through the dendrite to the left and right sides was assumed to be identical. Following this assumption and accounting for all synaptic inputs, a two-dimensional matrix was formed with axes corresponding to the time-versus-space dimensions: x vs. t. The resulting mesh contained numerical values representing the change in membrane potential caused by modeled excitatory synaptic inputs, both in time (from the moment of the first input, occurring at 0.5 ms after t=0 and in space (with the location of first input at t=0). 

Modeling and most calculations were carried out in MATLAB R2020a and KaleidaGraph software (v5), and the results were visualized using the “surface” function with the “hsv” colormap.

### 4.7. Key Model Parameters and Availability of Code

Key parameters of the model are summarized in Table 1.

## 5. Conclusions

Employing a plausible dendritic model, this study elucidates the role of exNMDA in influencing signal propagation along dendritic structures. The investigation delves into the optimal parameters governing input signals and exNMDA for the maximization of excitation propagation along the dendritic arbor. Nonetheless, the precise mechanism underlying excitatory signal amplification by exNMDA remains unverified and characterized by a paucity of comprehensive understanding. An in-depth exploration of this phenomenon is imperative, as it promises to augment our comprehension of the computational capacities intrinsic to neurons and dendrites. Furthermore, such inquiry extends to elucidating the intricate interplay between glial cells and neurons, particularly in the context of neurobiological processes such as learning.

## Figures and Tables

**Figure 1 ijms-25-04235-f001:**
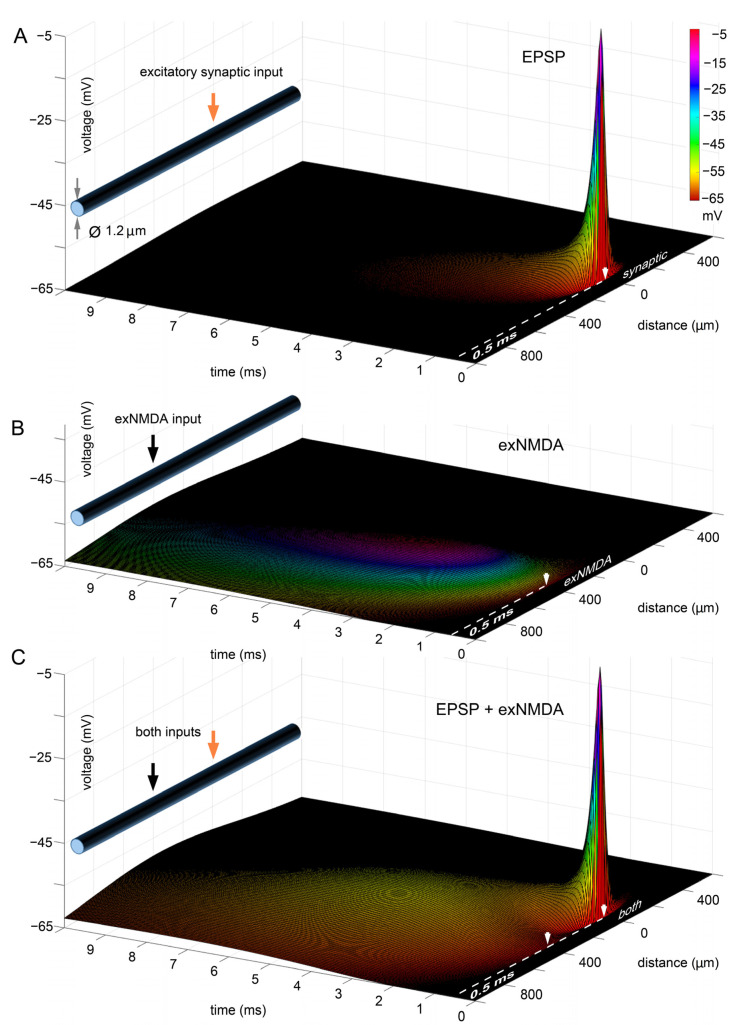
Separate and combined effects of single-synaptic and extra-synaptic inputs in a model of dendritic shaft. For modeling parameters, see “Methods”. (**A**), AMPA and NMDA receptor single synaptic stimulation at time point of 0.5 ms and the distance point of 0 µm or 1000 µm from the farther edge of the distance axis; (**B**) activation of extra-synaptic (exNMDA) receptors, located at the distance of 400 µm from point 0 and at time point of 0.5 ms; (**C**) simultaneous delivery of ligand to both synaptic receptors and exNMDA. (**A**–**C**) White arrows indicate position and initiation time for synaptic input or/and exNMDA cluster.

**Figure 2 ijms-25-04235-f002:**
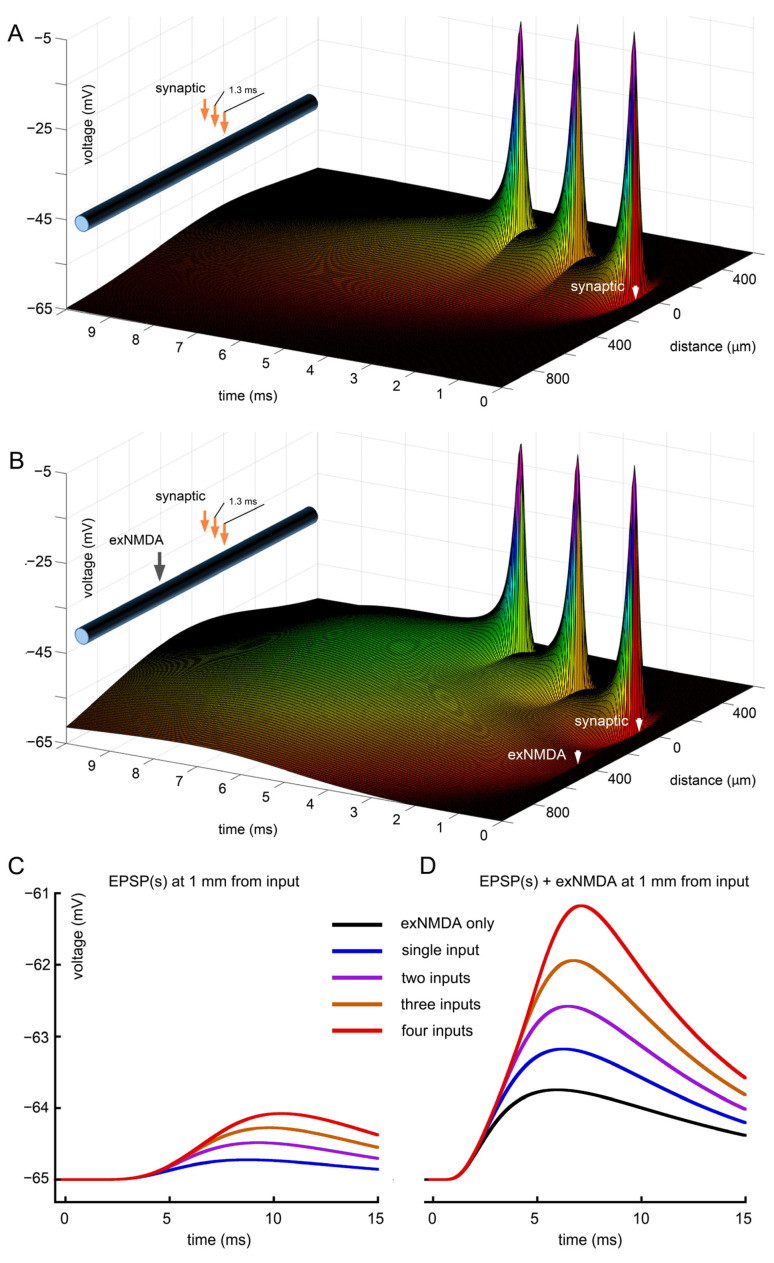
Separate and combined effects of triple synaptic and extra-synaptic inputs. (**A**) Sequential stimulation of 3 synapses starting from the time point of 0.5 ms and 1.3 ms apart at neighboring locations separated by 1 µm from each other; (**B**) activation of synaptic (as in **A**) and extra-synaptic (exNMDA) receptors, located at the distance of 400 µm from 0 and at time point of 0.5 ms; (**C**) comparison of voltage traces at “soma” for synaptic stimulations alone (from one to four inputs); (**D**), comparison of voltage traces at “soma” for synaptic (as in (**C**)) and exNMDA co-activation. Black trace—exNMDA only. (**A**,**B**) White arrows indicate position and initiation time for synaptic input and exNMDA cluster.

**Figure 3 ijms-25-04235-f003:**
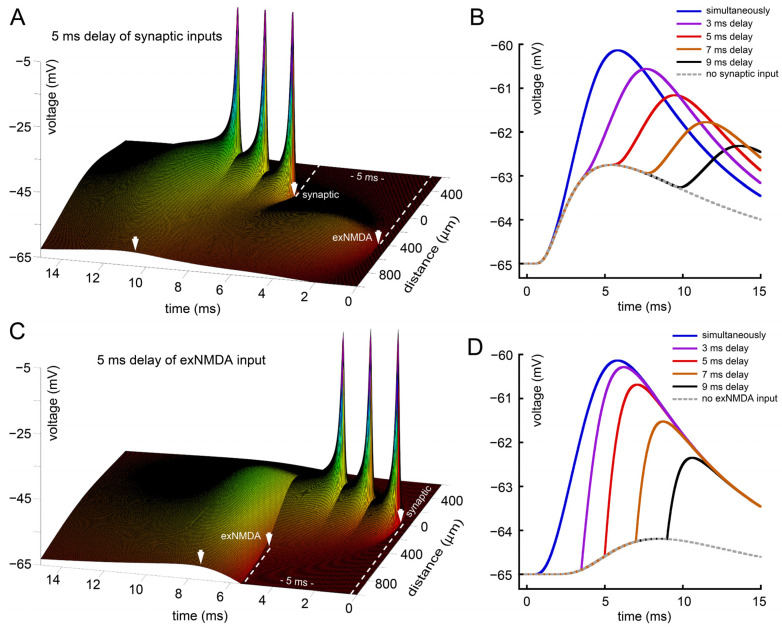
Effect of latency between synaptic and extra-synaptic inputs. (**A**) Three sequential synaptic stimuli (as in Figure 2) delayed by 5 ms after exNMDA; (**B**) comparison of voltage profiles at “soma” for synaptic inputs, delayed by 3–9 ms after exNMDA, dotted line—no synaptic input; (**C**) exNMDA delayed by 5 ms after synaptic inputs; (**D**) comparison of voltage profiles at “soma” for exNMDA activation, delayed by 3–9 ms after the first synaptic input, dotted line—no exNMDA. (**A**,**C**) White arrow on a time axis indicates peak value of membrane potential at the “soma” side. White arrows on a distance axis indicate position and initiation time for synaptic input and exNMDA cluster.

**Figure 4 ijms-25-04235-f004:**
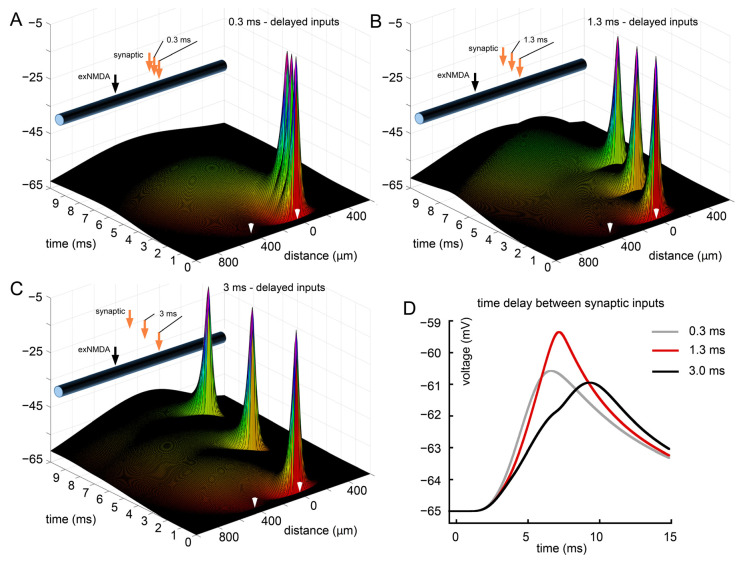
Analysis of the influence of synaptic stimulation frequency. (**A**) Three sequential synaptic stimuli separated by 0.3 ms and combined with exNMDA; (**B**) three sequential synaptic stimuli separated by 1.3 ms and combined with exNMDA; (**C**) three sequential synaptic stimuli separated by 3 ms and combined with exNMDA; (**D**) comparison of voltage traces at “soma” calculated for three synaptic inputs, differently distributed in time, and co-activated with exNMDA. (**A**–**C**) White arrows indicate position and initiation time of synaptic input (right) and exNMDA (left).

**Figure 5 ijms-25-04235-f005:**
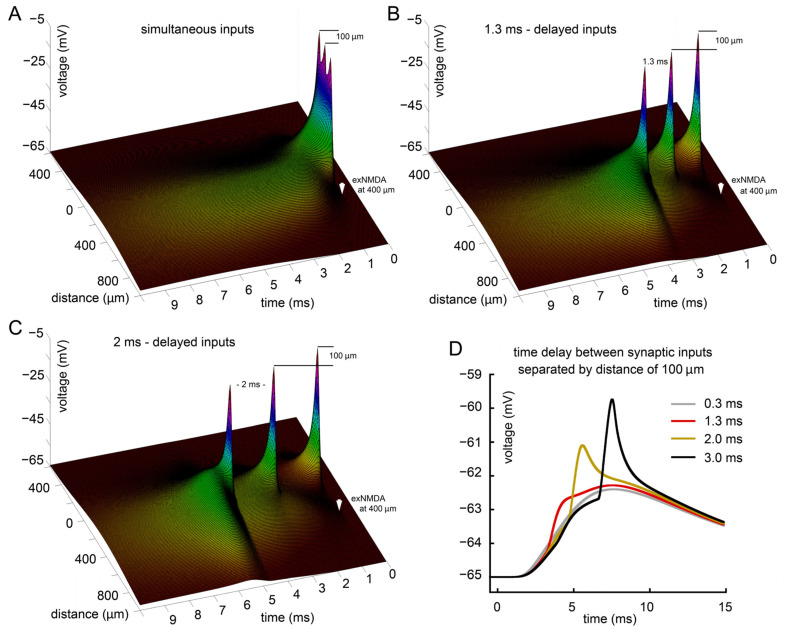
Effect of spatiotemporal distribution of EPSPs on the interaction with exNMDA. (**A**) Three simultaneous synaptic stimuli separated by the distance of 100 μm and combined with exNMDA; (**B**) three sequential synaptic stimuli separated by 1.3 ms and by the distance of 100 μm; (**C**) three sequential synaptic stimuli separated by 2 ms and by the distance of 100 μm; (**D**) comparison of voltage traces at far edge of the dendrite (“soma”) calculated for three synaptic inputs, separated by various time delays (from 0.3 to 3 ms) and by the distance of 100 μm, and co-activated with exNMDA. (**A**–**C**) White arrow indicates position and initiation time of exNMDA.

**Figure 6 ijms-25-04235-f006:**
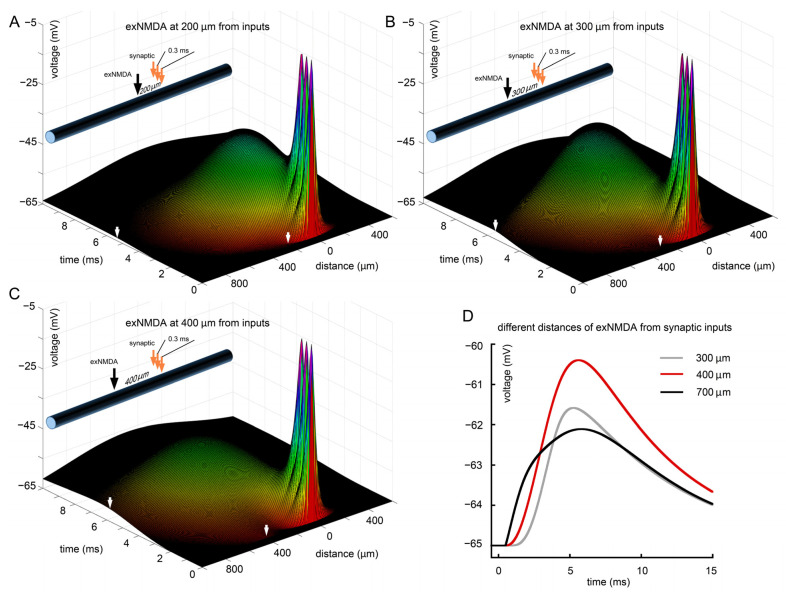
The optimal location of exNMDA. (**A**) Three sequential synaptic stimuli co-activated with exNMDA, located at 200 μm from EPSPs; (**B**) three sequential synaptic stimuli co-activated with exNMDA, located at 300 μm from EPSPs; (**C**) three sequential synaptic stimuli co-activated with exNMDA, located at 400 μm from EPSPs; (**D**) comparison of voltage traces at “soma” calculated for three synaptic inputs, co-activated with exNMDA, located at various distances from EPSPs (300, 400, and 700 μm). (**A**–**C**) White arrow on a distance axis indicates position and initiation time of exNMDA. White arrow on a time axis means peak value of membrane potential at the “soma” side.

**Figure 7 ijms-25-04235-f007:**
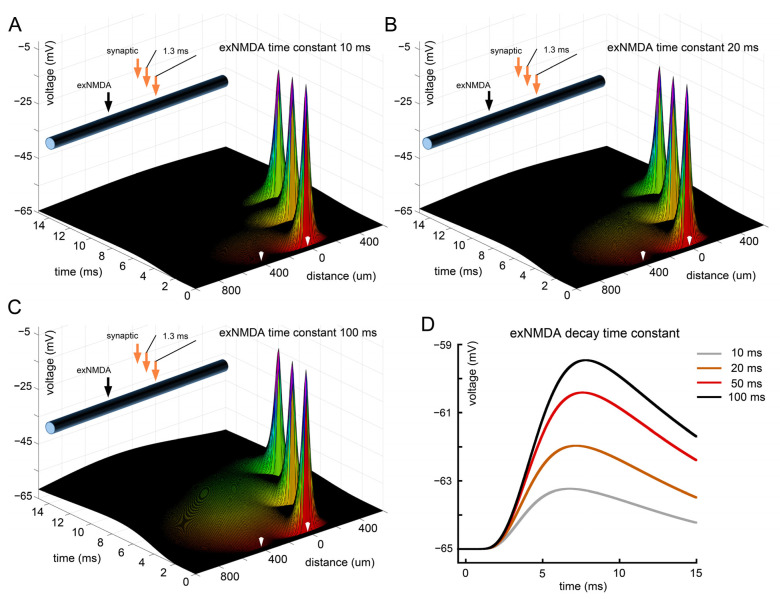
Effect of exNMDA decay time constant on the efficiency of interaction with EPSPs. (**A**) Three sequential synaptic stimuli, co-activated with exNMDA, characterized by decay time constant of 10 ms; (**B**) three sequential synaptic stimuli, co-activated with exNMDA, characterized by decay time constant of 20 ms; (**C**) three sequential synaptic stimuli, co-activated with exNMDA, characterized by decay time constant of 100 ms; (**D**) comparison of voltage traces at “soma”, calculated for three synaptic inputs, co-activated with exNMDA, characterized by various decay time constants (10, 20, 50, and 100 ms). (**A**–**C**) White arrows indicate position and initiation time of synaptic input (right) and exNMDA (left).

**Figure 8 ijms-25-04235-f008:**
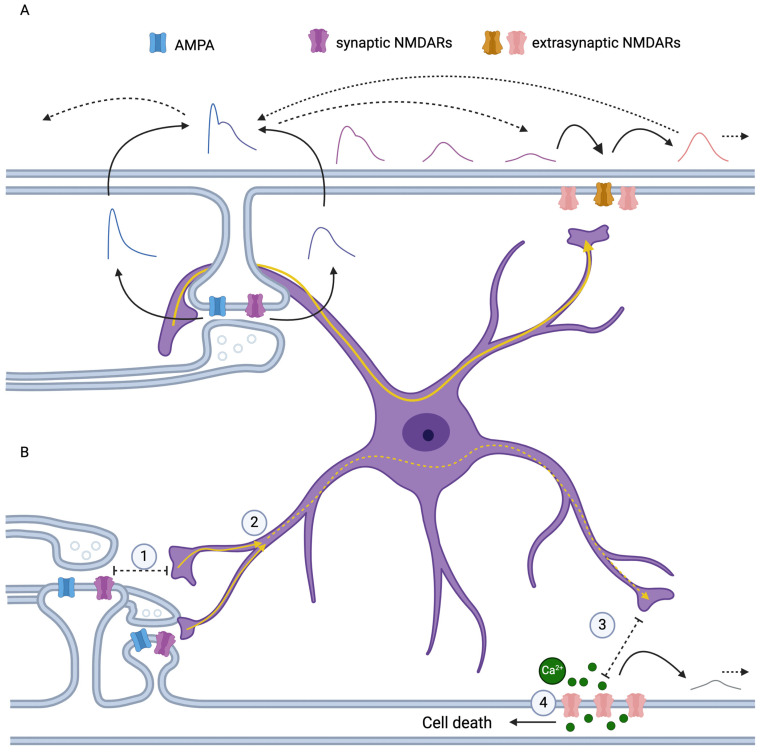
Scheme of the working model. (**A**) AMPA- and NMDA-receptor-induced depolarization in synapse creates a wave propagating to the exNMDA cluster. Signal is amplified by exNMDA cluster. Arrows with solid lines correspond to causal relationship. Dotted lines mean propagation of signal along the dendrite. Solid yellow lines mean Ca^2+^ signal spreading through astrocyte. (**B**). Extrasynaptic NMDA receptors are affected by some source of glutamate. The most probable source is astrocyte. Successful work of exNMDA signal amplification may depend on astrocyte health. (1) Increased distance between astrocyte leaflet and synaptic cleft may cause the decrease of sensing of synaptic events in the particular astrocyte and (2) cause deficient Ca^2+^ in compartment for initialization Ca^2+^ wave spreading from bunch of synapses to the related exNMDA cluster. (3) Impairment of astrocyte–exNMDA communication or balancing the distance between them leads to excessive of deficient gliotrnsmission causing inadequate signaling amplification. (4) Changes in exNMDA subunit ratio with increased NR2B or overstimulation of cluster may lead to increased Ca^2+^ influx and related cell damage. The resulting amplitude is low and ineffective when reaching the soma. Dotted yellow lines mean impaired Ca^2+^ signaling in astrocyte. Possible impact of astrocytes on exNMDA is not addressed by model.

**Table 1 ijms-25-04235-t001:** Key parameters of the model. Presented parameters are selected from the range of values taken from literature. *—calculated parameters. **—assumed parameters. ***—technical parameters.

Description	Value	References
$\mathrm{Resting}\;\mathrm{membrane}\;\mathrm{potential},\;V_{m}$	−65 mV	
$\mathrm{Equilibrium}\;(\mathrm{reversal})\;\mathrm{synaptic}\;\mathrm{potential},\;V_{eq}$	0 mV	
$\mathrm{Leakage}\;\mathrm{equilibrium}\;\mathrm{potential},\;E_{L}$	−65 mV	
Number of synaptic AMPA receptors, n	20 units	[82,83]
$\mathrm{Maximum}\;\mathrm{conductance}\;\mathrm{of}\;a\;\sin\mathrm{gle}\;\mathrm{AMPA}\;\mathrm{receptor},\;g_{AMPA}$	30 pS	[1,2,84]
$\mathrm{Maximum}\;\mathrm{conductance}\;\mathrm{of}\;\mathrm{AMPA}\;\mathrm{receptors},\;G_{AMPA}$	250 pS	*
AMPA receptor opening probability, p	0.8	[85]
$\mathrm{Maximum}\;\mathrm{AMPA}\;\mathrm{current},\;i_{AMPA}$	16 pA	*
$\mathrm{AMPA}\;\mathrm{receptor}\;\mathrm{rise}\;\mathrm{time}\;\mathrm{constant},\;\tau_{rise}$	0.07 ms	[3,4]
$\mathrm{AMPA}\;\mathrm{receptor}\;\mathrm{decay}\;\mathrm{time}\;\mathrm{constant},\;\tau_{decay}$	0.7 ms	[3,4]
Number of synaptic NMDA receptors, n	12 units	[82,83]
$\mathrm{Maximum}\;\mathrm{conductance}\;\mathrm{of}\;a\;\sin\mathrm{gle}\;\mathrm{NMDA}\;\mathrm{receptor},\;g_{NMDA}$	50 pS	[81]
$\mathrm{Maximum}\;\mathrm{synaptic}\;\mathrm{NMDA}\;\mathrm{receptors}\;\mathrm{conductance},\;G_{NMDA}$	600 pS	*
$\mathrm{Maximum}\;\mathrm{synaptic}\;\mathrm{NMDA}\;\mathrm{receptor}\;\mathrm{current},\;i_{NMDA}$	6 pA	*
$\mathrm{NMDA}\;\mathrm{receptor}\;\mathrm{rise}\;\mathrm{time}\;\mathrm{constant},\;\tau_{rise}$	3 ms	[86]
$\mathrm{NMDA}\;\mathrm{receptor}\;\mathrm{decay}\;\mathrm{time}\;\mathrm{constant},\;\tau_{decay}$	20 ms	[5,6]
$\mathrm{NMDA}\;\mathrm{receptor}\;\mathrm{kinetic}\;\mathrm{equilibrium},\;K_{eq}$	3.6 mM	[83]
$\mathrm{Extracellular}\;\mathrm{magnesium}\;\mathrm{concentration},\;[{Mg}^{2+}{]}_{ex}$	1.2 mM	[83]
Number of extrasynaptic NMDA receptors, n	80 units	[28]
$\mathrm{Extrasynaptic}\;\mathrm{NMDA}\;\mathrm{receptor}\;\mathrm{decay}\;\mathrm{time}\;\mathrm{constant},\;\tau_{decay}$	50–100 ms	[5,6]
$\mathrm{Maximum}\;\mathrm{synaptic}\;\mathrm{current},\;i_{syn}$	18 pA	*
Synaptic input frequency, f	100 Hz	[87]
$\mathrm{Input}\;\mathrm{resistance}\;\mathrm{of}\;\mathrm{synaptic}\;\mathrm{domain},\;r_{m}$	50 GΩ	[78,88]
Dendrite radius, r	0.6 μm	**
Dendrite time constant, τ	15 ms	*
$\mathrm{Specific}\;\mathrm{axial}\;\mathrm{resistance},\;R_{a}$	100 Ω×cm	[89]
Dendritic space constant, λ	400 μm	[75]
Membrane λ $\;\mathrm{resistance},\;r_{\lambda }$	1.85 GΩ	[75]
Time step resolution, t	10 μs	***
Space step resolution, x	0.1 μm	***

The code upon which the model is based is written in MATLAB with some features to solve each specific problem and is available upon request.

## Data Availability

The code is available at: https://www.dropbox.com/scl/fi/oww3xz7tyoupuu5bt7rcp/How-to-use-the-Public-folder.rtf?rlkey=efx61vt9rah0gmvmzqyb1ksep&dl=0, accessed on 9 April 2024.

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
