# Peer review of "Computational Modeling of Extrasynaptic NMDA Receptors: Insights into Dendritic Signal Amplification Mechanisms"

_ijms, 2024, doi:10.3390/ijms25084235_

Round 1
Reviewer 1 Report (Previous Reviewer 3)
Comments and Suggestions for Authors
I have no further comments
Author Response
Dear Reviewer 1,
Thank you very much for reviewing our manuscript. We appreciate your time and effort.
Kind regards,
Authors
Reviewer 2 Report (New Reviewer)
Comments and Suggestions for Authors
The effect of extrasynaptic NMDA receptors (exNMDA) on dendritic signal processing was discussed by the authors in this manuscript. MATLAB R2020a and KaleidaGraph software were used for the modeling and computations, and the "surface" function and "hsv" colormap were used for visualization. They discovered that a cluster of about 80 exNMDA receptors contributes to signal propagation, with extrasynaptic NMDA receptors having slower decay times and higher conductance than synaptic receptors. Overall, this work highlights the role of exNMDA receptors in neural information processing and sheds light on their functional significance in improving signal transmission within neurons. Since the topics the authors analyze are very hard to perform in the lab with conventional patch clamp techniques, model simulation represents a novel approach in this field. However, authors need to carefully set various parameters and give thorough explanations of their rationality in order to get as close as possible to real neurophysiological phenomena. Since I'm not an expert in this area (model simulation), my questions about this are limited to a few minor points.
1. In terms of the modeling portion, the authors did not go into great detail; they just stated that MATLAB R2020a and KaleidaGraph software were used for computation and visualization. Specifically, it would be beneficial for future researchers to be able to replicate and validate the authors' simulation results under controlled conditions if the MATLAB code were made available as supplemental data.
2. What is the methodology used in part 2.7 to determine each of the "Key Model Parameters and Availability of Code"? Specifically, why are there 20 synaptic AMPA receptors and 12 synaptic NMDA receptors, respectively,? Please provide an explanation based on the physiological background.
3. Amidst the diverse outcomes, I fail to comprehend the rationale behind the exNMDA receptor's 400 μm distance setting. I believe that the synaptic and extrasynaptic NMDA receptors that influence EPSP formation should be found in the same neuron. Could the writers clarify why they only modeled exNMDAR at 400 μm and not at other distances?
4. Minor: All figures should have "μ" instead of "u" in the "u" letters.
Author Response
Reviewer 2
Thank you very much for your thoughtful and helpful comments, which helped us to improve the quality of the manuscript, making it clearer and more complete.
- In terms of the modeling portion, the authors did not go into great detail; they just stated that MATLAB R2020a and KaleidaGraph software were used for computation and visualization. Specifically, it would be beneficial for future researchers to be able to replicate and validate the authors' simulation results under controlled conditions if the MATLAB code were made available as supplemental data.
Definitely, the main code will be provided as supplementary material and all the variations will be made public.
- What is the methodology used in part 2.7 to determine each of the "Key Model Parameters and Availability of Code"? Specifically, why are there 20 synaptic AMPA receptors and 12 synaptic NMDA receptors, respectively,? Please provide an explanation based on the physiological background.
Thank you for the comment. The parameters for the model are consistent with data from the literature. However, the range of values for certain parameters can vary significantly, whereas the model requires one single value. So, for example, why were the values chosen 20 synaptic AMPA receptors and 12 synaptic NMDA receptors in synapses? The problem is that synapses are very different and it cannot be said that there are average values for all synapses. For example, cerebellar neurons may contain more than 100 AMPA receptors [Tanaka, J. I., et al. (2005). Number and density of AMPA receptors in single synapses in immature cerebellum. Journal of Neuroscience, 25(4), 799-807]. In our case, we selected values adequate for cortical and hippocampal neurons [Li, S., Raychaudhuri, S. et al. (2021). Asynchronous release sites align with NMDA receptors in mouse hippocampal synapses. Nature communications, 12(1), 677; Chater, T. E., & Goda, Y. (2022). The shaping of AMPA receptor surface distribution by neuronal activity. Frontiers in synaptic neuroscience, 14, 833782].
Other parameters are computed using known parameters and these computations are demonstrated in methods. To make it clear we added an additional column “references” to the Table 1 with key parameter of the model. If value was received by computation we marked it by *, which is noted now below the table.
- Amidst the diverse outcomes, I fail to comprehend the rationale behind the exNMDA receptor's 400 μm distance setting. I believe that the synaptic and extrasynaptic NMDA receptors that influence EPSP formation should be found in the same neuron. Could the writers clarify why they only modeled exNMDAR at 400 μm and not at other distances?
In our study, we aimed to investigate the optimal positioning of extrasynaptic NMDA receptors (exNMDAR) relative to synaptic inputs on the same neuron's dendrite. Our goal was to understand how the spatial arrangement between these components influences signal processing. To address this, we introduced Section 3.5, "The Optimal Location of exNMDA" (Figure 6), in the Results chapter. This section delves into the specific distances between synaptic inputs and exNMDA clusters. Our simulations reveal that under our defined parameters, an optimal distance of 400 µm from the synaptic inputs to the exNMDA clusters leads to the most effective signal amplification. It's important to note that this optimal distance is contingent upon the length of the dendritic segment (in our case, assumed to be 1 mm from inputs to the soma). Varying dendritic parameters would likely result in different optimal distances, as the goal was to demonstrate the existence of an optimal location rather than a fixed universal distance.
- Minor: All figures should have "μ" instead of "u" in the "u" letters.
Thank you. Fixed.
Reviewer 3 Report (New Reviewer)
Comments and Suggestions for Authors
Makarov et al. investigated the role of extrasynaptic NMDA receptors in the propagation of synaptic signals along dendrites by using a computational model of dendrites. This is an interesting paper elucidating the intricate interplay between glial cells and neurons, particularly in the context of neurobiological processes such as learning. The paper is well written. There are a few issues to be addressed to further improve the manuscript.
1. The authors should add in the title that the present study is a computational model study.
2. Introduction is too long and redundant. It should be more concise.
3. In figure legend of figure3, B and C are interchanged.
Author Response
Dear reviewer 3,
Thank you for your careful and useful evaluation of our manuscript, which helped us to improve its quality.
- The authors should add in the title that the present study is a computational model study.
Thank you for this suggestion. We have changed a title to “Computational Modeling of Extrasynaptic NMDA Receptors: Insights into Dendritic Signal Amplification Mechanisms”.
- Introduction is too long and redundant. It should be more concise.
We completely agree. For now, introduction is much shorter. We replaced a part with review of existing exNMDA models to discussion. From lines 121-167 to 640-686.
- In figure legend of figure3, B and C are interchanged.
Thank you for careful reading. Fixed. Line 428.
Reviewer 4 Report (New Reviewer)
Comments and Suggestions for Authors
The role of extrasynaptic NMDA receptors in the propagation of synaptic signals along dendrites
The premise of this paper is interesting and needed as it bring attention to this issue of extrasynaptic receptors in general as well as specifically the NMDA type.
The implications of this study within neural networks is worth considering for publications even if one only addressed this with computational analysis. Empirical data could address this concept with Ca2+ indicators and antibody staining of the receptor locations. I am sure experimentalist will like this paper to justify experiments to address this concept.
The manuscript was a please to read. The logical flow of the manuscript was nicely performed and explanations explained the modeling outcomes well.
I did not check the math or if the equations were valid. They seem logical to address the questions being addressed.
I do not have any edits to suggest for the authors.
It appears you do not need the “Informed Consent Statement:” at the end of this manuscript.
Author Response
Dear Reviewer 4,
Thank you very much for reviewing our manuscript. We appreciate your time and effort.
Kind regards,
Authors
Reviewer 5 Report (New Reviewer)
Comments and Suggestions for Authors
The work presents the modeling approach for studying the role of exNMDA clusters for passively conducting electrical signal propagation along the dendritic structure. The model predicts the optimal parameters for single cluster exNMDA, providing the most effective amplfication, including its position and decay constant. Given the limitation of respective experimental setup, the model provides the powerful tool for exploring parameter space with minimal cost.
Minor comments: 1) for parameters in Table 1 it would be good to have the source column (references), where these numbers were taken from
2) Is it possible to evaluate the robustness of obtained solutions? Something like local sensitivity to to test what happens if parameter values vary a little bit
3) How long does it take for simulation of a single scenario? (in MATLAB)
Author Response
Reviewer 5
Thank you very much for your thoughtful and helpful comments, which helped us to improve the quality of the manuscript, making it clearer and more complete.
Minor comments:
1) for parameters in Table 1 it would be good to have the source column (references), where these numbers were taken from
Thank you for this suggestion. We added a column with references to the Table 1.
2) Is it possible to evaluate the robustness of obtained solutions? Something like local sensitivity to to test what happens if parameter values vary a little bit.
Definitely. We have simulated a model with different parameters to provide validation. Some of presented experimental data such as time delay between synaptic inputs (Figure 4) or different number of inputs (Figure 2) may be counted as validation. In addition, we tested the model with different parameters to assess the reliability of the solutions. Moderate changes in the number of receptors, rise time, or decay constant for synaptic AMPA or NMDA did not produce significant changes. Most importantly, changing the dendrite diameter had the greatest impact on the results, which is consistent with the cable theory. The code will be provided as supplementary material and all the variations of validation tests will be made public.
3) How long does it take for simulation of a single scenario? (in MATLAB)
Thank you for your question. A single scenario simulation took about 3 minutes.
This manuscript is a resubmission of an earlier submission. The following is a list of the peer review reports and author responses from that submission.
Round 1
Reviewer 1 Report
Comments and Suggestions for Authors
The paper is well-written and easy to understand, even for those without a strong background in neuroscience. The paper effectively introduces the role of dendrites, and NMDA receptors and clearly outlines the specific research question of how distal exNMDA clusters amplify signals. The paper emphasizes the importance of spatial positioning and NR2B subunits for signal amplification.
Suggestions:
- While the paper mentions the focus on near exNMDA receptors, a more nuanced discussion of the existing literature and its limitations would strengthen the introduction.
- The paper relies solely on a computational model. Including in vitro or in vivo experiments (if applicable) would strengthen the findings and increase generalizability.
- Briefly discussing or acknowledging alternative mechanisms for signal amplification beyond exNMDA clusters would provide a more balanced perspective.
- Briefly outlining the key features and assumptions of the computational model would be helpful for readers unfamiliar with the approach.
- The paper mentions future studies but could benefit from more specific suggestions for future research directions based on the findings.
Overall, this paper presents a well-written investigation into the role of distal exNMDA clusters in signal amplification using a computational model.
Author Response
Thank you for your careful and useful evaluation of our manuscript, which helped us a lot to improve its quality.
- While the paper mentions the focus on near exNMDA receptors, a more nuanced discussion of the existing literature and its limitations would strengthen the introduction.
Done. We have added and expanded the discussion of the possible role of exNMDA receptors. However, in our opinion, an attempt to review existing literature to a certain extent dilutes the main focus of the manuscript.
- The paper relies solely on a computational model. Including in vitro or in vivo experiments (if applicable) would strengthen the findings and increase generalizability.
Thank you for this suggestion. Indeed, it would be much better to strengthen or even replace the model with real experimental data. Unfortunately, to our best knowledge there are very limited and technically unrealizable approaches to address the role of exNMDA without affecting the synaptic ones. In the discussion, we included a paragraph on the limitations for experimental testing of the hypothesis.
- Briefly discussing or acknowledging alternative mechanisms for signal amplification beyond exNMDA clusters would provide a more balanced perspective.
We completely agree. Now in introduction and discussion sections, we have mentioned NMDA spikes to rightly point out that alternative mechanisms of signal amplification exist and how they differ from our proposed way of amplification. We also think it is important to emphasize that NMDA spikes leave little or no room for signal summation, since these spikes operate on an all-or-nothing basis, whereas our proposed alternative only slightly amplifies the signals, but does not negate the importance of space-time summation and fine dendritic computation.
- Briefly outlining the key features and assumptions of the computational model would be helpful for readers unfamiliar with the approach.
Done. We have now added a table with key parameters.
- The paper mentions future studies but could benefit from more specific suggestions for future research directions based on the findings.
We agree. We have now discussed some possible future experiments and their associated technical limitations.
Reviewer 2 Report
Comments and Suggestions for Authors
Reviewer comments
Dendrites can be thought of as analogous to transistors in a computer, performing simple operations using electrical signals. Dendrites contain voltage-gated ion channels, giving them the ability to generate action potentials. Extra-synaptic NMDA channels stand out as one of the main contributors to these channels. This study highlights how crucial the ideal spatial placement of the NMDA cluster is to influencing signal amplification efficiency. This work deviates from the norm by focusing on the relationship between distal exNMDA (> 100 μm) and synaptic inputs, which are arranged in clusters that serve as signal amplifiers.
Recommendation
1. The paper is scientifically sound, well-planned, and written in an organized manner.
2. The research objective and work targets are original and relevant to the field. The outcomes of the study are discussed briefly.
3. The schematic representation of results is nice.
4. Material and methods are described well.
5. References are appropriate.
Scientific comment
1. The discussion is very short and is not well-established with previously published papers.
2. NMDAR receptors are linked with various neuro diseases including AD, HD, Stroke, and IBI. Hyperactivity or hypofunction of NMDARs possibly contributes to neurodisease pathophysiology. Discuss briefly how this study result helps to overcome these diseases.
3. Provide a future perspective and application of your study.
Minor comment/s
1. Figure 8 is not very informative and can be improved by some additions.
2. In line 153, check the word 'about 2 μm2'.
3. Check the reference pattern and rewrite it accordingly.
Author Response
Thank you very much for your thoughtful and helpful comments, which helped us improve the quality of the manuscript, making it clearer and more complete.
Scientific comments
- The discussion is very short and is not well-established with previously published papers.
We agree. We have now added two important paragraphs to the discussion. The first concerns hypothetical mechanisms of pathology based on the theoretical data of our model. This will make the manuscript better suited to the theme of the special issue (“Novel Hypotheses for Dementia and Neurodegenerative Diseases: From Molecular Mechanisms to Therapies”). In the second important addition to the text, we discuss possible experimental refinements of our hypothesis and their technical limitations.
- NMDAR receptors are linked with various neuro diseases including AD, HD, Stroke, and IBI. Hyperactivity or hypofunction of NMDARs possibly contributes to neurodisease pathophysiology. Discuss briefly how this study result helps to overcome these diseases.
We think, it is a very important and useful comment. Now a discussion on contribution to neurodegenerative disease is added.
- Provide a future perspective and application of your study.
Done. A paragraph on possible experiments and their limitation is now added to discussion.
Minor comments
- Figure 8 is not very informative and can be improved by some additions.
We have improved Figure 8 to make it more descriptive and to relate it to functional impairment/disease states. We have also added to the discussion a section on the hypothetical contribution of the described mechanism to pathogenesis.
- In line 153, check the word 'about 2 μm2'.
Done
- Check the reference pattern and rewrite it accordingly.
Done. We have checked and corrected reference list.
Reviewer 3 Report
Comments and Suggestions for Authors
-
In this manuscript, Makarov et al. aimed to build computational models to understand how exNMDA receptors can amplify distal signals. Here are my suggestions to improve it.
-
The English writing needs substantial improvement. Here are several examples, but I can't exhaust all of them.
- Line 222-223: "…the effect of dendritic thickness? I think the authors mean the tapering of dendrites or irregular diameter?"
- Line 249: "What does a single synaptic input mean?"
- Line 257: "Decays at a distance of 400 um. To what degree? The model has no soma/AIS."
- Line 150: "Consists of about 0 mV?"
- There are a few typos, such as:
- "Is equation 2 correct?"
- Line 205: "AMPA receptors?"
- Line 226: "Ra = 100 ohm cm?"
-
This model is not realistic; please tone it down.
-
Line 246: As a theoretical paper, the code should be public on websites such as ModelDB.
-
Line 45: Not all dendritic propagation relies on passive propagation. See dendritic spikes observed in Purkinje neurons and pyramidal neurons such as https://doi.org/10.1016/j.celrep.2018.07.011; https://doi.org/10.1523/JNEUROSCI.1719-20.2020; https://doi.org/10.1523%2FJNEUROSCI.1717-07.2007
-
. The authors should cite relevant papers and make it clear that some dendrites can have dendritic spikes. This story only applies to passive dendrites
needs improving
Author Response
Thank you for careful evaluation of our manuscript, which helped us to improve its quality and correct some technical mistakes.
- The English writing needs substantial improvement. Here are several examples, but I can't exhaust all of them.
Line 222-223: "…the effect of dendritic thickness? I think the authors mean the tapering of dendrites or irregular diameter?"
Right, “thickness” has been replaced by “diameter” and additional clarification added.
Line 249: "What does a single synaptic input mean?"
Now we have clarified this point in the text.
Line 257: "Decays at a distance of 400 um. To what degree? The model has no soma/AIS."
This is a good point. Definitely, we do not have in the model cell body as а separate compartment. We conventionally called corresponding to “soma” the signal level that reaches the far “end” of the simulated cable, where the body of a large pyramidal cell of the cortex or hippocampus may begin. The 400 µm parameter was mentioned as a length constant, but we agree with the reviewer that this is not self-evident and have adjusted this piece of text accordingly.
Line 150: "Consists of about 0 mV?"
We have clarified this point in the text referring to the equilibrium potential of excitatory synapse.
Overall, we have revised the text and substantially improved the English, and consulted on some expressions with a native English speaker.
- There are a few typos, such as:
"Is equation 2 correct?"
We carefully checked all the equations. “t-1” component of the formula refers to the voltage at the previous time point.
Line 205: "AMPA receptors?"
Fixed, thank you.
Line 226: "Ra = 100 ohm cm?"
Fixed, thank you.
- This model is not realistic; please tone it down.
We agree. “Realistic” has been toned down to “plausible”.
- Line 246: As a theoretical paper, the code should be public on websites such as ModelDB.
Definitely, the main code will be provided as supplementary material and all the variations will be made public.
- Line 45: Not all dendritic propagation relies on passive propagation. See dendritic spikes observed in Purkinje neurons and pyramidal neurons such as https://doi.org/10.1016/j.celrep.2018.07.011; https://doi.org/10.1523/JNEUROSCI.1719-20.2020; https://doi.org/10.1523%2FJNEUROSCI.1717-07.2007
(https://doi.org/10.1016/j.celrep.2018.07.011; https://doi.org/10.1523/JNEUROSCI.1719-20.2020; https://doi.org/10.1523%2FJNEUROSCI.1717-07.2007)
Thanks for the comment. We completely agree with this point. Now in the introduction we point to NMDA spikes as well as voltage-gated propagation as alternative mechanisms for conducting and amplifying signals in the dendrites of several types of neurons. The references mentioned are now included in the list.
- The authors should cite relevant papers and make it clear that some dendrites can have dendritic spikes. This story only applies to passive dendrites.
Done (please, see previous comment).
Round 2
Reviewer 1 Report
Comments and Suggestions for Authors
The author has made significant revisions to the manuscript, addressing the feedback provided. The revised manuscript is now considered acceptable for publication.
Reviewer 3 Report
Comments and Suggestions for Authors
The authors solved my concerns.
Comments on the Quality of English Languagenow ok